# Chemogenetic Depletion of Hypophysiotropic GnRH Neurons Does Not Affect Fertility in Mature Female Zebrafish

**DOI:** 10.3390/ijms23105596

**Published:** 2022-05-17

**Authors:** Sakura Tanaka, Nilli Zmora, Berta Levavi-Sivan, Yonathan Zohar

**Affiliations:** 1Department of Marine Biotechnology, Institute of Marine and Environmental Technology, University of Maryland Baltimore County, Baltimore, MD 21202, USA; stanaka@umbc.edu (S.T.); nzmora@umbc.edu (N.Z.); 2Department of Animal Sciences, The Robert H. Smith Faculty of Agriculture, Food, and Environment, The Hebrew University of Jerusalem, Rehovot 76100, Israel; berta.sivan@mail.huji.ac.il

**Keywords:** GnRH, LH, ovulation, zebrafish, fertility

## Abstract

The hypophysiotropic gonadotropin-releasing hormone (GnRH) and its neurons are crucial for vertebrate reproduction, primarily in regulating luteinizing hormone (LH) secretion and ovulation. However, in zebrafish, which lack GnRH1, and instead possess GnRH3 as the hypophysiotropic form, GnRH3 gene knockout did not affect reproduction. However, early-stage ablation of all GnRH3 neurons causes infertility in females, implicating GnRH3 neurons, rather than GnRH3 peptides in female reproduction. To determine the role of GnRH3 neurons in the reproduction of adult females, a *Tg(gnrh3:Gal4ff; UAS:nfsb-mCherry)* line was generated to facilitate a chemogenetic conditional ablation of GnRH3 neurons. Following ablation, there was a reduction of preoptic area GnRH3 neurons by an average of 85.3%, which was associated with reduced pituitary projections and *gnrh3* mRNA levels. However, plasma LH levels were unaffected, and the ablated females displayed normal reproductive capacity. There was no correlation between the number of remaining GnRH3 neurons and reproductive performance. Though it is possible that the few remaining GnRH3 neurons can still induce an LH surge, our findings are consistent with the idea that GnRH and its neurons are likely dispensable for LH surge in zebrafish. Altogether, our results resurrected questions regarding the functional homology of the hypophysiotropic GnRH1 and GnRH3 in controlling ovulation.

## 1. Introduction

Reproduction in vertebrates is driven by the hypothalamus–pituitary–gonadal (HPG) axis, in which internal and external cues are translated into endocrine signals that control reproductive performance. The hypophysiotropic gonadotropin-releasing hormone (GnRH), a neuropeptide found in specialized neurons in the hypothalamic preoptic area (POA) of all studied vertebrates, is a crucial regulator of the reproductive HPG axis. GnRH induces the synthesis and secretion of the two gonadotropins, follicle-stimulating hormone (FSH) and luteinizing hormone (LH), from the pituitary gonadotropes. In mice and humans, natural homozygous mutations in the *GNRH1* gene result in hypogonadal hypogonadism and infertility due to a significant decrease in FSH and LH circulating levels [1,2,3]. Furthermore, the inherited depletion of GnRH neurons in female mice leads to infertility and failure to ovulate due to an inability of the few existing GnRH neurons to induce the required LH surge [4]. In teleosts, the loss of the hypophysiotropic GnRH in female medaka (*gnrh1*^−/−^) resulted in infertility attributed to an inadequate LH surge required for ovulation [5].

Most vertebrates possess two or three forms of GnRH paralogs [6,7,8]. As described above, the predominant hypophysiotropic variant, GnRH1, is a non-conserved and species-specific isoform, and is the primary regulator of reproduction [9]. Its neurons, located in the POA of the hypothalamus [10,11], innervate the median eminence of mammals, or directly in the pituitary (in teleosts) to stimulate pituitary gonadotropes [12]. GnRH2 is conserved, and the most ubiquitous isoform of GnRH in all vertebrates except for rodents and jawless fishes, and its neurons are found in the midbrain tegmentum [13]. GnRH3 is a teleost-specific conserved isoform [14]. Its neurons mainly populate the olfactory bulbs/terminal nerve (OB/TN), and hypothalamus. Some teleosts lost the GnRH1 gene during evolution of the *Ostariophysi*/cypriniform lineage [6], and feature only two isoforms: GnRH2 and GnRH3. In these species, including the cyprinid family which includes the zebrafish (*Danio rerio*), GnRH3 and its neurons are believed to have adopted the functions of the hypophysiotropic GnRH1, which include the regulation of FSH and LH secretions from the pituitary for both gonadal development and maturation/ovulation [7,8,15]. 

It is commonly accepted that GnRH1 and GnRH3 are derived from gene duplication of a common ancestor, which is supported by the similarity in phylogeny of GnRH1 and GnRH3, as well as in neuronal origin and developmental migration patterns [16]. In addition, functional homology between the hypophysiotropic GnRH1 and GnRH3 of zebrafish neurons as regulators of reproduction is mainly derived from their hypothalamic neuroanatomical distributions in the preoptic area (POA) [17,18,19,20], and their projections to the pituitary in adults [21]. Furthermore, laser ablation of GnRH3 neurons at 4- and 6-days post-fertilization (dpf) resulted in arrested oocyte development and infertility in female zebrafish [22], underscoring the importance of GnRH3 neurons in reproduction, especially in ovulation. However, GnRH3 knockout zebrafish (*gnrh3*^−/−^) notably exhibit normal fertility, including no effects on ovulation [23,24], thus implying that gametogenesis and the pre-ovulatory LH secretion from the pituitary gonadotropes commence independently of hypophysiotropic GnRH signaling. These findings may be explained by compensatory mechanism(s) that mitigate for the lack of GnRH3 functions in *gnrh3*^−/−^ and/or the presence of a functional redundancy, which are manifested in the *gnrh3*^−/−^, but not in the GnRH3 neuron-ablated zebrafish. Compensation and redundancy, if they exist, can originate from different neurons along the hypothalamus–pituitary (HP) axis or reside autonomously in GnRH3 neurons themselves, and can be fulfilled by structurally homologous factors [25,26,27,28,29], neurotransmitters [30,31], or by other factors that intrinsically induce adaptation [32]. The option of autonomous action by GnRH3 neurons is supported by the deleterious effect on oocyte development following early-stage laser ablation of GnRH3 neurons [22].

GnRH3 neurons in zebrafish, such as GnRH1 neurons, display a specific ontogenic development that includes a well-documented and conserved migration pattern from the nasal placode to their destination in the POA of the hypothalamus [17,18,19,20]. In mammals, any interference of GnRH1 neuron maturation, migration, or GnRH1 secretion causes the failure of puberty, fertility, and other reproductive functions [33,34,35,36,37,38]. Similarly, the absence of GnRH3 neurons may impair the development of the reproductive HPG axis (e.g., pituitary gonadotropes differentiation). Therefore, when studying any impact of the hypophysiotropic GnRH in relation to ovulation, it is imperative to eliminate the neurons only after the HPG axis is well and fully developed, preferably prior to the time of the pre-ovulatory LH surge.

To determine if GnRH3 neurons are required for the pre-ovulatory LH surge and ovulation, we generated a transgenic zebrafish, allowing for conditional ablation of only GnRH3 neurons, and eliminated an average of 85.3% of GnRH3 neurons in reproductively cycling female zebrafish by using the chemogenetic nitroreductase (NTR)-mediated cell ablation method. Bacterial NTRs are enzymes that enable the conversion of non-toxic prodrugs, e.g., metronidazole (Mtz), into toxic metabolites, inducing conditional cytotoxicity, specifically in cells expressing NTR [39]. *Escherichia coli*-derived NTR has been successfully implemented in zebrafish with the combination of Mtz to ablate specific cells/tissues without side effects [40,41,42]. We then examined the effect of GnRH3 neuron chemical ablation on LH secretion and reproductive competence. Our findings revealed that GnRH3 and its neurons are likely not essential for inducing the pre-ovulatory LH surge in this species, supporting the hypothesis that the potential redundancy and/or compensation originates in other locations, not in GnRH3 neurons along the HP axis.

## 2. Results

### 2.1. Evaluation of Transgenic Line Tg(gnrh3:Gal4ff; UAS:nfsb-mCherry)

To evaluate the generated *Tg(gnrh3:Gal4ff; UAS:nfsb-mCherry)* transgenic zebrafish females expressing NTR-mCherry fusion protein specifically in GnRH3 neurons, we used double-immunohistochemistry to assess the co-localization of NTR-mCherry in GnRH3 neurons. Whole brains that were co-immunostained using GnRH3-GnRH-associated protein (GAP) and mCherry antibodies revealed that an average of 90.5% of the detectable POA GnRH3 neurons also expressed the transgene, NTR-mCherry (Appendix A). The number of double-labeled neurons varied, and 5 of the 14 examined females displayed 100% co-localization, another 5 had over 90%, and the remaining 4 females displayed less than 90% co-localization (68−83%).

### 2.2. Attenuation of GnRH3 Neurons Is Verified

GnRH3 neuronal ablation was induced by Mtz exposure in sexually mature transgenic females for a total of 28 days. The exposure regime of 14 h light and 10 h dark, in which 2 mM Mtz was present only during the dark period, was optimized to avoid side effects that could compromise fish health, while ensuring efficiency of the treatments (Figure 1A). Confocal images of intact brains of Mtz-treated transgenic fish at 14 days and 28 days of treatment confirmed the depletion of GnRH3 neurons (Figure 1B).

The reduction in the number of GnRH3 neurons and GnRH3 induced by the Mtz treatment in the transgenic fish was validated by double immunostaining of GnRH3-GAP and mCherry antibodies in whole brains, quantifying *gnrh3* mRNA levels in whole brains, and immunostaining of GnRH3-GAP in the pituitaries to detect GnRH3 neuronal projections. The experimental groups consisted of one test group of *Tg(gnrh3:Gal4ff; UAS:nfsb-mCherry)* fish exposed to Mtz, and two control groups: *Tg(gnrh3: Gal4ff; UAS: nfsb-mCherry)* fish exposed only to vehicle, and WT siblings, which did not express NTR-mCherry, treated with Mtz.

Double immunostaining of the whole brains revealed that the number of GnRH3-immunoreactive (ir) cells in the POA was significantly reduced in the GnRH3 neuron-ablated group (Tg-Mtz), having 2 ± 0.5 (mean ± SEM) remaining GnRH3-ir cells, which is roughly an 85.3% reduction of the control groups, Tg-Cont (14.7 ± 1 positive cells, *p* = 1 × 10^−^^7^), and WT-Mtz (12.7 ± 0.9 positive cells, *p* = 1 × 10^−8^) (Figure 1C,D). The number of mCherry-ir cells in the POA also significantly declined by around 10 times in Tg-Mtz to 1.2 ± 0.4 positive cells, as compared to the Tg-Cont group (13.5 ± 0.6 positive cells, *p* = 3.58 × 10^−9^) (Figure 1C,D). 

When the relative *gnrh3* mRNA levels in the GnRH3 neuron-ablated group (Tg-Mtz) was compared with those of the two control groups, Tg-Mtz fish displayed a significant decrease by 10 times of the WT-Mtg (*p* = 0.01), and 5 times of the Tg-Cont (*p* = 0.04) groups (Figure 1E). Next, we immunostained the whole pituitary using GnRH3-GAP antibody, confirming the reduction of GnRH3 neuronal projections in GnRH3 neuron-ablated fish (Tg-Mtz) compared to the control groups (Figure 2A).

### 2.3. Plasma LH Levels Are Unchanged in GnRH3 Neuron-Ablated Females

To determine whether GnRH3 neuron ablation affects LH secretion from the pituitary, we measured plasma LH levels using a specific ELISA for LH after spawning at 28 days of treatment. Plasma LH levels of GnRH3 neuron-ablated females (Tg-Mtz) were unchanged compared to those of the control females (Tg-Cont and WT-Mtz; *p* = 0.6) (Figure 2B), which ranged from 215 to 1323 pg/mL.

### 2.4. GnRH3 Neuron-Ablated Females Exhibit Normal Fertility

To investigate the effect of GnRH3 neuronal ablation on spawning and fertility, we evaluated reproductive characteristics in each group at 14 and 28 days of treatment by allowing each experimental female to mate with a WT male counterpart. The first observation was that all females spawned within the allocated time. There was no significant difference in the number of fertilized eggs (fecundity), fertilization rate, or hatching rate between the three treatment groups during each trial (*p* = 0.489 for the number of fertilized eggs; *p* = 0.161 for fertilization rate; *p* = 0.456 for the hatching rate) (Figure 3A). Further, there was no difference in the gonadosomatic index (GSI) between the three treatment groups after spawning at 28 days of treatment (*p* = 0.9) (Figure 3B). The ovaries of GnRH3 neuron-ablated females (Tg-Mtz) possessed normal oocytes at all stages of development, and were undistinguishable from those of the two control groups (Tg-Cont and WT-Mtz) (Figure 3C). Importantly, no correlation was observed between the number of POA GnRH3-ir neurons and each reproductive assessment parameter at 28 days of treatment: fecundity (*p* = 0.846, *R* = 0.059), fertility (*p* = 0.7633, *R* = −0.085), and hatching rate (*p* = 0.2461, *R* = −0.3193) (Figure 4). This indicates that female fish with 1−2 detectable GnRH3 neurons in the POA performed similarly to the those with 16 and 19 neurons (Figure 4).

### 2.5. Pituitary GnRH2 Peptide Contents Are Unchanged in GnRH3 Neuron-Ablated Females

To examine whether the reduction of GnRH3 neurons induces an increase of the other zebrafish GnRH forms, GnRH2 in the pituitary and pituitary GnRH2 peptide content was measured using GnRH2 ELISA immediately after the spawning evaluation at 28 days of treatment. The reduction of GnRH3 neurons did not increase pituitary GnRH2 peptide contents (*p* = 0.6) (Appendix A).

### 2.6. Ablated GnRH3 Neurons Regenerate after the Removal of Mtz during Early Development

GnRH3 neuron ablation during early development was induced by 5 mM Mtz or 10 mM Mtz for 72 h, starting at 24 h post-fertilization (hpf); 10 mM Mtz resulted in complete disappearance of the NTR-mCherry labeled GnRH3 neurons (Appendix A). The recovery of GnRH3 neurons was observed at 24 h and 48 h after removal of the Mtz in the OB and POA (Appendix A). To answer whether an early prolonged treatment with Mtz completely and irreversibly ablates GnRH3 neurons, zebrafish were treated with 5 mM Mtz for 14 days starting at the 6 hpf blastula stage. One month after the removal of the Mtz, the brains were examined for signs of GnRH3 neuron regeneration (Appendix A). Though no signal was observed immediately after the treatment, GnRH3 neuron cell bodies recovered during the following month, and their neuronal projections were found throughout the entire brain (Appendix A).

### 2.7. Acute Ablation of GnRH3 Neurons in Adult Females Does Not Affect Fertility 

To investigate the effect of GnRH3 neuronal ablation on spawning and fertility, we treated sexually mature female *Tg(gnrh3:Gal4ff; UAS:nfsb-mCherry)* and WT siblings with 10 mM Mtz for 24 h (Appendix A), following a protocol described elsewhere [21]. We then evaluated the reproductive characteristics in each group one month after removal from the treatment water, by allowing each experimental female to mate with a WT male counterpart. There was no significant difference in the number of fertilized eggs (fecundity), fertilization rate, or hatching rate between the treatment groups before, at, or after the treatment (*p* = 0.576 for the number of fertilized eggs; *p* = 0.06 for fertilization rate; *p* = 0.296 for the hatching rate) (Appendix A). 

## 3. Discussion

The hypophysiotropic GnRH and its neurons are believed to be the ultimate regulators of reproduction by controlling LH and FSH secretion in all vertebrates. In most species, including mammals, reptiles, amphibians, and some teleosts such as Perciformes, the hypophysiotropic GnRH is the species-specific GnRH1 [9]. Elimination studies of GnRH1 or its neurons in both mammalian and teleost species have established indispensability with respect to reproductive fitness, predominantly in female ovulation [1,2,3,4,5] (teleost: Figure 5A). Since the hypophysiotropic form in zebrafish is GnRH3 [6], we sought to understand its exact roles in zebrafish reproduction, and compare its functions to those of GnRH1. We previously eliminated the GnRH3 gene via TALEN gene knockout [23] and, in previous study, completely abolished GnRH3 neurons by targeted laser ablation [22]. To our surprise, though complete GnRH3 neuronal ablation starting at early developmental stages (4- and 6-dpf) resulted in infertile adult females [22], the inherited lack of hypophysiotropic GnRH3 in zebrafish (*gnrh3*^−/−^) resulted in fertile individuals [23,24]. These contradicting outcomes raised the question of whether GnRH3 peptide function in reproduction differs from that of the GnRH1 peptide. Specifically, it called into question whether GnRH3, similar to GnRH1, plays a role in inducing the preovulatory LH surge in adult females. In order to investigate whether GnRH3 neurons are crucial in the regulation of ovulation and spawning, we conditionally ablated GnRH3 neurons in sexually mature female zebrafish using NTR-mediated chemogenetic cell ablation, and assessed reproductive performance. Surprisingly, like in targeted GnRH3 gene knockout [23,24], a dramatic reduction in the number of POA GnRH3 neurons in these females resulted in normal fertility, despite notable decreases in brain *gnrh3* mRNA levels and pituitary GnRH3 neuronal projections. Furthermore, there was no correlation in reproductive performance between females having 1–4 remaining POA GnRH3 neurons and females with 10–19 neurons. At first examination, our results indicate that, unlike other species which display GnRH1-dependent reproduction, the hypophysiotropic GnRH and its neurons are likely dispensable for the regulation of maturation, ovulation, and spawning in female zebrafish.

On the other hand, using the NTR-mediated cell ablation method, a previous experiment showed that *Tg(gnrh3:Gal4ff; UAS:nfsb-mCherry)* females that were exposed to 5 mM Mtz for 24 h exhibited infertility 1 month after the treatment [21]. The discrepancy with our results, despite using the same *Tg(gnrh3:Gal4ff)* line source and the same conditions, is possibly due to the difference in the transgene expression levels, or the higher percentage of GnRH3 neurons expressing the transgene. If true, it can be postulated that some level of GnRH3 neuron presence/activity, even if it is minute, is sufficient for ovulation and spawning in zebrafish females. The fact that zebrafish and other species have not lost the GnRH3 gene during evolution suggests that GnRH3 and its neurons maintain some indispensable roles, such as the neuromodulation of sexual behavior [43,44].

In the discussion that follows, we incorporate our past and current results with other relevant findings to delineate possible scenarios by which reproduction is controlled in female zebrafish, with special attention to the role of GnRH3 and its neurons in the process (Figure 5).

### 3.1. GnRH1 and GnRH3 Functionally Differ: GnRH3 Likely Functions as a Neuromodulator and Not as an LH Secretagogue

Despite the resemblance between GnRH3 and GnRH1 peptides and their neurons, and though the loss of the hypophysiotropic GnRH1 in medaka (*gnrh1*^−/−^) results in infertility in females [5], *gnrh3*^−/−^ zebrafish maintain normal fertility [23,24]. Since the *gnrh1*^−/−^ medaka females had fully developed oocytes but failed to ovulate, it can be postulated that GnRH3 in zebrafish, despite being hypophysiotropic, does not play a role in inducing the pre-ovulatory LH secretion. However, our current results contradict the results from the study in which laser ablation of GnRH3 neurons at 4- and 6-dpf zebrafish resulted in arrested oocyte development and infertility in female zebrafish [22]. Though these results suggest that GnRH3 and its neurons maintain key roles in reproduction [21,22] (Figure 5A), several alternative scenarios may explain this contradiction. One is that GnRH3 plays an important role only in the development of the reproductive HP axis before puberty (hence the arrested oocytes in the early GnRH3 neuron ablation study). A second scenario is that the loss of GnRH1 during the evolution in the *Ostariophysi*/cypriniform lineage [6] triggered the assignment of other factors in the regulation of LH secretion and ovulation, resulting in functional redundancy (Figure 5B). In that regard, GnRH3 has been known as a neuromodulator of sexual behavior [43,44]. Therefore, as a consequence of neofunctionalization [45], GnRH3 functions are perhaps confined to neuromodulation rather than to LH secretion. 

In mammals, any interference of GnRH1 neuronal migration or GnRH1 secretion causes failure to attain puberty, fertility, and other reproductive functions [33,34,35,36,37,38]. In the present study, despite being successfully ablated during the migration phase, GnRH3 neurons were able to fully recover within 14 days, or 1 month after the removal of Mtz. This phenomenon did not allow us to examine the effects of GnRH3 neuron depletion on the development of the HPG axis. It is possible that the laser-based method used to ablate GnRH3 neurons in zebrafish larvae also damaged the neighboring progenitors or cells involved in the migratory tract of the GnRH3 neurons. This may have hampered the ability of GnRH3 neurons to regenerate, whereas the chemogenic ablation is more specific and confined to the GnRH3 neurons. If true, this may explain why chemically ablated GnRH3 neurons regenerated. The only study in zebrafish with a permanent elimination of a GnRH3 migratory factor is a recent study where prokinetic receptor was knocked out (*prokr1b*^−/−^), resulting in alternation in GnRH3 neuronal migration of the rostral part and at the level of the anterior commissure. However, *prokr1b*^−/−^ fish display normal gonadal development and normal fertility [46]. Based on these results, it seems that the altered migration and projection pattern of GnRH3 neurons does not compromise reproduction. Assuming that GnRH3 projections to the pituitary are hampered in the *prokr1b*^−/−^ fish, these results support a non-hypophysiotropic role for GnRH3. Therefore, the most logical explanation that can explain the results deriving from the disrupted GnRH3 neuronal migration, as well as their ablation at early development and at adulthood, is that GnRH3 neurons have an important neuromodulatory role during development (and not in adult reproduction) that does not require accurate distribution of GnRH3 projections. 

### 3.2. The Effect of the Hypophysiotropic GnRH on Pituitary LH Secretion and lhb Expression Is Still Controversial in Female Zebrafish

Zebrafish are batch spawners, and mRNA levels of female pituitary *lhb*, a gene encoding the beta subunit of LH, increase starting 2 h after lights-on, and remain high throughout the rest of the daylight phase [47], presumably to support maturation and ovulation of the next batch of ovulating oocytes. Therefore, the similar plasma LH levels between the GnRH3 neuron-ablated females and the control fish after spawning likely reflects the lack of distinct differences during the pre-ovulatory LH surge. Since *lhb* mutant female zebrafish display normal gonadal development, but not oocyte maturation and ovulation [48], LH signaling is also critical to oocyte maturation and ovulation in zebrafish. Thus, our current results that GnRH3 neuron-ablated females display normal fertility indicates that the significant reduction of GnRH3 neurons does not affect LH signaling in regulating oocyte maturation and ovulation. 

As to the question of whether GnRH3 elicits LH secretion in zebrafish, a recent study reported that in vitro pituitary incubation with GnRH3 alone does not affect LH release from the pituitary [49]. Similarly, in vivo intracerebroventricular injection with GnRH3 has no effect on plasma LH levels in sexually mature zebrafish females [49]. Another group has also shown that intraperitoneal (IP) injection of long-acting GnRH1 analog does not induce a statistically significant increase in plasma LH levels, though there is a slight increase [50]. One of the reasons for the inconsistency may be due, in part, to relatively low affinity to the receptor, or to changing temporal expression levels of the GnRH receptor, *gnrhr2*, in the pituitary [47,51]. Furthermore, it is known that in zebrafish and other cypriniform fish (e.g., common carp and goldfish), LH release is strongly inhibited by dopamine [52,53,54]. Considering that dopamine D2 receptors are expressed in LH zebrafish gonadotropes [54], the fluctuating expression of dopamine D2 receptors may cause the inconsistent responses to GnRH3 signaling on LH gonadotropes in zebrafish. For example, treatment with GnRH3 in a primary culture of zebrafish pituitary cells and GnRH3 IP injection with domperidone, a D2 receptor antagonist, in sexually mature or regressed females persistently upregulated the expression of pituitary *lhb* [49,54,55]. Overall, the sum of the pharmacological effects of GnRH3 on LH secretion and *lhb* mRNA levels from multiple studies in zebrafish is inconsistent and debatable. Taken together with our ablation results, it may be postulated that GnRH3 does not have a prominent hypophysiotropic role in adult zebrafish like that of GnRH1 in other vertebrate species (Figure 5C).

### 3.3. Other Central and Peripheral Factors May Induce LH Secretion in Sexually Mature Female Zebrafish 

The possibility that redundant factors function in the absence of GnRH3 to maintain normal reproduction has been suggested recently [56]. To date, various neuropeptides, such as kisspeptin, neurokinin B, secretoneurin, and vasoactive intestinal peptide (Vip), were shown to stimulate LH secretion in zebrafish [49,50,57,58]. However, knockout of the above factors in zebrafish, except for secretoneurin, yielded fertile fish, ruling them out as paramount LH secretagogues.

Generally, a high concentration of estrogen produced by the follicles triggers LH surge and subsequently induces final oocyte maturation and ovulation, which is referred to as positive feedback. In mammals, kisspeptin neurons in the anteroventral periventricular nucleus or POA that innervate POA GnRH1 neurons, whose kisspeptin expression is positively regulated by gonadal steroid feedback, induce a GnRH1/LH surge when the estrogen circulation level is high, and ultimately induce ovulation (see the referenced review [59]). On the contrary, targeted mutations of the two kisspeptin and their receptors together with GnRH3 result in fertile medaka and zebrafish [24,60,61]. Therefore, unlike mammals, teleosts may utilize distinct estrogen positive feedback pathways to induce LH surge. Therefore, steroid modulating roles to induce ovulation are still arguable in teleosts [62]. Since medaka *gnrh1*^−/−^ females are infertile [5], gonadal estrogen signals may be mediated via GnRH1 at the brain level to regulate LH surge in this species. Though it is still possible that in zebrafish, gonadal estrogen signals are mediated through GnRH3 neurons even under a significant depletion of GnRH3 neurons, a previous study reported that estradiol-17*b* (E_2_) directly acts on the pituitary to induce LHb protein synthesis in pubertal female zebrafish [63]. Congruently, in vitro pituitary incubation demonstrated that E_2_ directly induces LH secretion from mature female zebrafish pituitaries [49]. These facts indicate that direct pituitary–gonadal estrogen feedback mechanisms possibly exist in zebrafish (Figure 5D), whereas physiological evidence for brain source regulation of ovulation in vivo is still lacking. These findings support the idea that GnRH3 is not required to induce the LH surge in female zebrafish (Figure 5B,C).

Though the suggested scenario applies to females at the prime of their reproductive phase, it may not apply to aging female fish. Aging, characterized by depleted levels of GnRH in mammals, leads to depleted LH levels and lack of ovulation [64]. Moreover, the ablation of GnRH3 neurons may reflect natural GnRH depletion in aged zebrafish females (over 2-years-old). In fact, Vip was shown to rescue the observed GnRH depletion only in aged female rats [65]. In addition to the direct innervation of GnRH neurons by Vip neurons in the suprachiasmatic nucleus, it was recently demonstrated that Vip neurons innervate kisspeptin neurons in the anteroventral periventricular and rostral periventricular nuclei, and the arcuate nucleus of the hypothalamus of mice, to modulate the negative steroid feedback [66]. Therefore, the depletion in GnRH3 neurons in zebrafish females may be mitigated by the induction of LH directly by Vip [49], through kisspeptin neurons or via other unknown factors.

### 3.4. A Small Amount of GnRH3 May Still Trigger LH Secretion 

The possibility that only a few hypophysiotropic GnRH3 neurons are sufficient to induce LH secretion from pituitary gonadotropes should be taken into consideration. We observed that, after Mtz treatment of the *Tg(gnrh3:Gal4ff; UAS:nfsb-mCherry)* fish, some of the remaining GnRH3 neurons expressed NTR, whereas some did not. In that regard, it has been reported by several studies that stable transgenic lines frequently show variegated or diminished expression over time [67]. The number of POA GnRH3 neurons was significantly reduced (to 14.7%) in GnRH3 neuron-ablated fish as compared to the control groups, and is associated with a dramatic decrease (10 times) relative to *gnrh3* mRNA levels. Such a reduction in the *gnrh3* mRNA level is similar in ratio to the relative *gnrh3* mRNA levels in *gnrh3*^−/−^ zebrafish embryos reported elsewhere [68]. Although there is no statistical significance in the relative *gnrh3* mRNA levels between the two control groups, Tg-Cont and WT-Mtz, their relative mRNA expression levels varied in relation to the ablated females (Tg-Cont: 5 times; WT-Mtz; 10 times). This may be explained by potential background effects of the NTR expressed in GnRH3 neurons, even in the absence of Mtz. Our current results differ from previous studies in mice, where homozygous natural mutation in *GnRH1* resulted in hypogonadal hypogonadism caused by insufficient LH and FSH circulation, which led to infertility in all individuals [1,2,3]. Similarly, an 88% reduction of GnRH1 neurons in a GnRH1 mutant mouse (GNR23) strain failed to produce normal litters, estrous cycles, or ovulation in females [4,69]. A reduction of GnRH1 neurons in female GNR23 mice by 12% did not affect the onset of puberty, but rather impaired ovulation, likely due to the inability to generate an LH surge [4]. Similar to zebrafish and medaka, where males are not affected by the lack of hypophysiotropic GnRH functions [5,23], male GNR23 mice with 85% less GnRH1 neurons display normal fertility [69]. These findings indicate that simple pulsatile GnRH secretion is likely maintained with only a very small number of GnRH neurons [4]; hence, our results that GnRH3 neuron-ablated adult female (14.7% remaining POA GnRH3 neurons) fertility may indicate that the remaining few neurons in the POA can secrete sufficient GnRH3 to support the LH surge required for ovulation in zebrafish females. Unlike in mammals, there is no median eminence–pituitary portal blood system in the teleost and hypothalamic-derived axons that penetrate the neurohypophysis, and often directly innervate the pituitary cells [12]. This physiological phenomenon may enable minute levels of hypophysiotropic factors (e.g., GnRH3) to exert their functional influence on the pituitary gonadotropes to secrete LH and FSH (Figure 5E). Alternatively, it is possible that the significant reduction in the number of GnRH3 neurons in GnRH3 neuron-ablated zebrafish may cause an increase in neural activities of the remaining few GnRH3 neurons, leading to a typical LH secretion level (Figure 5E).

### 3.5. GnRH2 Likely Does Not Play a Role in Inducing Ovulation and Spawning in GnRH3 Neuron-Ablated Females 

Finally, pituitary GnRH2 protein content is unchanged in GnRH3 neuron-ablated females, suggesting that GnRH2 did not assume the role of GnRH3 in LH secretion. This result can be supported by the findings that female GnRH2 mutants (*gnrh2*^−/−^) and female mutants that lack all GnRHs (*gnrh2*^−/−^; *gnrh3*^−/−^) also maintain normal fertility in zebrafish [70,71]. In WT zebrafish, the number of GnRH2 neurons projecting to the pituitaries and reaching the LH gonadotropes are lower relative to those of GnRH3 neurons [70]. Therefore, unlike what is reported after fasting [70], the reduction in the number of GnRH3 neuronal projections to the pituitary may not provoke the plasticity of GnRH2 neurons.

### 3.6. Summary

Overall, our results demonstrate that the hypophysiotropic GnRH and its neurons are dispensable for the pre-ovulatory LH surge, and for ovulation in female zebrafish. Thus, the possibility that the LH surge and ovulation are regulated by redundant pathways is still feasible. Though further studies in other teleost species are necessary to firmly validate this idea, these findings propose that GnRH3, unlike GnRH1 in other vertebrates, is not a sole or key factor in inducing ovulation.

## 4. Material and Methods

### 4.1. Animal Husbandry

All zebrafish were maintained in a 28 °C recirculating system with a 14-h light/10-h dark cycle, and fed twice daily with commercial pellets (Gemma Micro 300; Skretting, North Tooele, UT, USA) and brine shrimp, *Artemia* sp. Tübingen WT was used for the experiments, including the generation of the *Tg(gnrh3:Galf4ff)* line. c264Tg: *Tg(UAS:nfsb-mCherry)* [72] was purchased by the Zebrafish International Resource Center (ZIRC, Eugene, OR, USA). GnRH3 mutants (*gnrh3*^−/−^) were generated in our previous study [23]. Before tissue dissection, adult zebrafish were deeply anesthetized using a concentration of >250 mg/L Tricaine (TCI America, Portland, OR, USA), and then sacrificed by decapitation. All animal experiments were performed in accordance with institutional guidelines, and were approved by the Institutional Animal Care and Use Committee at the University of Maryland School of Medicine (approval # 0519010).

### 4.2. Generation of Transgenic Fish

*Tg(gnrh3:Gal4ff; UAS:nfsb-mCherry)* double-transgenic fish expressing *nfsb* encoding the enzyme NTR and mCherry fluorescent fusion protein under the control of *gnrh3* promoter, which leads to specific expression in GnRH3 neurons, was generated by crossing *Tg(gnrh3:Gal4ff)* and *Tg(UAS:nfsb-mCherry)*. The *gnrh3*:*Gal4ff* construct, containing codon-optimized *Gal4ff* [73] under the zebrafish GnRH3 promoter [19], was kindly provided by Dr. Matan Golan (Institute of Animal Sciences, Agricultural Research Organization, Volcani Center, Israel) [21]. To generate *Tg(gnrh3:Gal4ff)* fish, the transgenic construct and *Tol2* transposase mRNA were co-microinjected into fertilized one-cell stage WT zebrafish eggs. Screening was performed using enhanced green fluorescent protein (eGFP) expression in the heart, since the *gnrh3*:*Gal4ff* construct also contained eGFP under the control of cardiac muscle-specific gene promoter (*my17:eGFP*). *Tg(gnrh3:Gal4ff)* fish were crossed with *Tg(UAS: nfsb-mCherry)* to generate *Tg(gnrh3:Gal4ff; UAS:nfsb-mCherry)* double-transgenic fish, and screened through mCherry expression in GnRH3 neurons. *Tg(gnrh3:Gal4ff; UAS:nfsb-mCherry)* females were raised to sexual maturity and mated with WT males. The collected embryos were separated into two groups based on mCherry expression in GnRH3 neurons, in which mCherry-positive fish were designated as *Tg(gnrh3:Gal4ff; UAS:nfsb-mCherry)* fish, and the mCherry-negative fish were designated as WT siblings of control fish, without *nfsb-mCherry* expression. 

### 4.3. Conditional Attenuation of GnRH3 Neurons in Adulthood

GnRH3 neuronal ablation was performed by the NTR-mediated cell ablation method, which is well developed in zebrafish [40,41,42]. Mtz (Sigma-Aldrich, St. Louis, MO, USA, Cat # M1547) was selected for use as a prodrug in this study, according to previous reports [40,74]. Three experimental groups were prepared: (1) Tg-Mtz: *Tg(gnrh3:Gal4ff; UAS:nfsb-mCherry)* fish with 2 mM Mtz in vehicle (0.2 % dimethyl sulfoxide (DMSO) in system water); (2) Tg-Cont: *Tg(gnrh3:Gal4ff; UAS:nfsb-mCherry)* fish with vehicle; and (3) WT-Mtz: WT siblings, without *nfsb-mCherry* expression, with 2 mM Mtz in vehicle. The treatments were performed for a total of 28 days in 9.46 L glass containers in a closed water system containing 2.5 L of treatment water. Each female was individually tagged through a specific fin clip pattern in the tail fins, thus allowing each fish’s reproductive performance to be tracked. The treatments were employed only during the 10-h dark period to prevent Mtz photodegradation, and the fish were maintained with an accurate reproductive circadian cycle. The treatment waters were freshly prepared each time, and twice-daily replaced after feeding the fish to apparent satiation. Most previously reported studies carried out the treatment at early developmental stages with relatively shorter exposure duration (24–48 h) using a 5–10 mM Mtz concentration (e.g., [75,76,77,78]). Considering the size of the adult zebrafish brain, as well as a previous example that utilized this method in adult fish [74], our preliminary trials indicated that a 14-day exposure to 2 mM Mtz is required to induce sufficient GnRH3 neuronal ablation in adult females without any off-target and deleterious health effects. Sexually mature females (5-months-old) were used for the treatment and further analysis. Two weeks prior to starting the treatment, transgenic or WT sibling female fish were mated with 5–8-months-old WT males, and only the successfully spawned females generating > 200 fertilized eggs and >80% fertilization rate, and their counterpart males were selected for the experiment. According to a previous study, we also exposed *Tg(gnrh3:Gal4ff; UAS:nfsb-mCherry)* fish and WT siblings to 10 mM Mtz for 24 h.

### 4.4. Conditional Attenuation of GnRH3 Neurons during Early Development

To induce GnRH3 neuronal ablation during early development, *Tg(gnrh3:Gal4ff; UAS:nfsb-mCherry)* larvae were exposed with 5 mM Mtz, 10 mM Mtz, or vehicle. All treatments were performed after dechorionation by incubating the embryos with 1 mg/mL Pronase (Roche, Basel, Switzerland) for a brief time. The treatment vehicle and system water contained 0.003% 1-phenyl-2-thiourea to prevent pigment formation. The treatment water was freshly prepared and replaced every 24 h throughout the duration of the experiment. The treatments were started at 24 hpf for a total of 72 h. On the other hand, 6 hpf blastula stage embryos were treated with 5 mM Mtz for 14 days. In both treatment conditions, the treated larvae were transferred to system water after completing the treatment window.

### 4.5. Characterization of Reproductive Performance

At 14 and 28 days of treatment, each experimental female was mated with a WT male. The female and male were separated by a plastic divider, which was removed immediately after lights-on at 0900 h, for 2-h to incite mating. Subsequently, all eggs were collected from each breeding chamber. The eggs were disinfected using 45 ppm sodium hypochlorite solution to prevent potential mortality by bacterial infection. The numbers of fertilized eggs and unfertilized eggs were counted to quantify fecundity and fertilization rate (number of fertilization eggs/total eggs oviposited). Hatching rate (number of hatched eggs/total fertilized eggs) was also calculated at 3-dpf. The experiment was repeated three times. Between the repetitions, the experimental fish treated with 10 mM Mtz for 24 h were allowed to recover in system water for 1 month. Finally, the reproductive parameters were examined by following the description above. 

### 4.6. Confocal Imaging 

Before the treatment, and at 14 and 28 days of treatment, GnRH3 neuron-ablated females (Tg-Mtz) were randomly selected, and the intact brains were removed via careful dissection, and mounted onto slides with 0.8% low-melting agarose. For the ablation of GnRH3 neurons during early development, larvae were randomly selected from each treatment group immediately before the treatment, and at 24 and 48 h (for larvae in the 72-h treatment group), and 1 month (for larvae in the 14-day treatment group) after removal from the treatment water. Randomly selected experimental larvae from each respective treatment group at each time interval were mounted onto slides containing 0.8% low-melting agarose. Confocal images were acquired from the ventral (adult brains) or dorsal (larvae) side using a Leica SP8 confocal microscope (Leica Microsystems, Wetzlar, Germany) to confirm GnRH3 neuronal ablation. The images were processed using Photoshop (Adobe, San Jose, CA, USA) and Fiji [79]. 

### 4.7. Tissue Sampling

After the spawning trial at 28 days of treatment (at 1100 h–1300 h; 2–4 h after lights on), blood samples were collected using a heparinized glass microcapillary syringe following the protocol described elsewhere [80]. Brains and pituitaries were dissected out from randomly selected fish from each respective experimental group, flash-frozen using dry ice, and stored at −80 °C until use. The remainder of the brains and pituitaries were fixed in 4% paraformaldehyde in phosphate-buffered saline (PBS) solution containing 0.25% TritonX-100 (PBT) overnight at 4 °C. Ovaries were sampled, and weights were recorded for gonadosomatic index calculations (GSI: gonad weight/body weight × 100), followed by fixation with Bouin’s solution (Sigma-Aldrich, St. Louis, MO, USA) to determine oocyte development and maturation status. 

### 4.8. Ovarian Histology

Randomly chosen ovaries fixed in Bouin’s solution were dehydrated with an ethanol series, and embedded in plastic resin using a JB-4 Embedding Kit (Sigma-Aldrich, St. Louis, MO, USA) according to the manufacturer’s protocol. Sections were prepared at 4 µm thickness, stained with Gill No. 2 hematoxylin and eosin-Y (Sigma-Aldrich, St. Louis, MO, USA), and subsequently mounted with DPX Mountant (Sigma-Aldrich, St. Louis, MO, USA). Images were photo-documented by using a Zeiss Axioplan microscope (Carl Zeiss Microscopy, Jena, Germany). The images were adjusted using Photoshop (Adobe, San Jose, CA, USA) and Fiji [79].

### 4.9. Real-Time Quantitative PCR

*gnrh3* mRNA levels in the pituitary after the treatment were measured by real-time quantitative PCR (qPCR). Total RNA was extracted from the dissected frozen brains using Trizol (Invitrogen, Waltham, MA, USA) according to the manufacturer’s protocol. An amount of 500 ng of total RNA was utilized for each 20 µL reaction of cDNA synthesis using a QuantiNova Reverse Transcription Kit (Qiagen, Hilden, Germany). Real-time qPCR was conducted by use of GoTaq qPCR Master Mix (Promega, Madison, WI, USA) with gene-specific primer sets (Appendix A) on a 7500 Fast Real-Time PCR System (Applied Biosystems, Waltham, MA, USA). The cycling condition included an initial denaturation of 95 °C for 2 min, followed by 40 repeating cycles of 95 °C denaturation for 3 s, and 60 °C annealing and extension for 30 s. The amplification specificity was verified by the single-peak melting curves produced by qPCR. Relative mRNA levels for *gnrh3* were determined in duplicate using the 2^−∆∆Ct^ method, and normalized against the level of the internal control gene, eukaryotic translation elongation factor 1 alpha (*eef1a1l1*; GenBank accession # NM_131263). 

### 4.10. Whole Mount Immunostaining

Fixed brains or pituitaries were washed with PBT and incubated in 150 mM Tris-HCl (pH 9.0) for 15 min at room temperature, followed by incubation for 20 min at 70 °C. The brains were then incubated in 0.05% Trypsin-EDTA (Gibco, Waltham, MA) for 1.5 h on ice and washed. The rinsed tissues were blocked with blocking buffer (2% normal goat serum; 1% BSA; 1% DMSO in PBT) for 1.5 h at room temperature, and incubated with rabbit anti-zebrafish GnRH3 GnRH-GAP antibody [23] and/or rat anti-red fluorescent protein (RFP) antibody (ChromoTek, Planegg-Martinsried, Germany, Cat # 5F8-20; RRID: AB_2336064) diluted 1:500 in antibody dilution buffer (1% BSA; 1% DMSO in PBT) overnight at 4 °C. After incubation with a secondary antibody or two secondary antibodies, Alexa Fluor 488-conjugated goat anti-rabbit IgG (Invitrogen, Waltham, MA, USA, Cat # A-11008; RRID: AB_143165) and/or Alexa Fluor 647-conjugated anti-rat IgG (Invitrogen, Waltham, MA, USA, Cat # A32795; RRID: AB_2762835) diluted 1:850 in antibody dilution buffer at 4 °C overnight, the tissues were washed with PBS and mounted onto slides with 0.8% low-melting agarose. The images were photo-documented from the ventral side using a Leica SP8 confocal microscope (Leica Microsystems, Wetzlar, Germany) with the setting, z-stack (201 ± 32 µm for the brains; 40 µm for the pituitaries). The number of GnRH3 immune-positive signals in the 3-dimensional images of the brain POAs were counted and recorded. The images were processed using Photoshop (Adobe, San Jose, CA, USA) and Fiji [79].

### 4.11. Protein Quantification by ELISA

Plasma LH levels were measured using carp LH ELISA [50,81] with some modifications based on a previous study [49]. For sample preparation to quantify individual pituitary GnRH2 peptide content, each frozen pituitary was sonicated in 200 µL of sterile H_2_O for 20 s utilizing a Sonifier 450 (Branson, Brookfield, CT, USA) with output control set at 2 and constant duty cycle. Of the 200 µL sonicated solution, 15 µL was used for the total protein measurement of each pituitary using a CBQCA Protein Quantitation Kit (Invitrogen, Waltham, MA, USA) following the manufacturer’s manual. Subsequently, 185 µL of 4 N acetic acid was added to the remaining sonicated solution, vortexed, and then left on ice for 15 min. After centrifugation at 14,000 rpm for 30 min, supernatants were collected in new tubes and lyophilized. Post-lyophilization, samples were resuspended with 60 µL of GnRH ELISA assay buffer (400 mM NaCl; 1 mM EDTA; 0.1% BSA; 0.001% sodium azide in 0.1 M phosphate buffer, adjusted to a pH of 7.4). The GnRH2 ELISA protocol has been described elsewhere [82]. The quantified GnRH2 peptide contents were normalized against the total pituitary protein contents of each sample and represented as GnRH2 peptide per 1 µg total pituitary protein. 

### 4.12. Statistical Analysis

The data in bar and line charts represent mean ± SEM. In box plots, the middle lines delineate median values, and the top and bottom lines indicate upper and lower quartiles. Rhombuses and circles represent mean and individual data points, respectively. R 4.1.1 software was used for all statistical analyses [83]. Data normality was validated using a Shapiro–Wilk test. For comparison between the two groups, a two-sample Student’s *t* test was performed. A one-way analysis of variance (ANOVA) test was performed for comparison among more than two groups. If the *p*-value was less than 0.05, a Tukey’s honest significant difference *post hoc* test was performed for pairwise multiple comparison. For analysis of reproductive performance results, a two-way mixed design ANOVA was performed to test differences between the treatment group. The statistical significance of the correlations between the number of POA GnRH3-ir neurons and each reproductive parameter was determined by calculating Pearson’s correlation coefficient, except for comparisons to hatching rate, which were tested by calculating Kendall’s tau correlation coefficient, since the data did not appear to be normally distributed. *p* < 0.05 was judged to be statistically significant in this study.

## Figures and Tables

**Figure 1 ijms-23-05596-f001:**
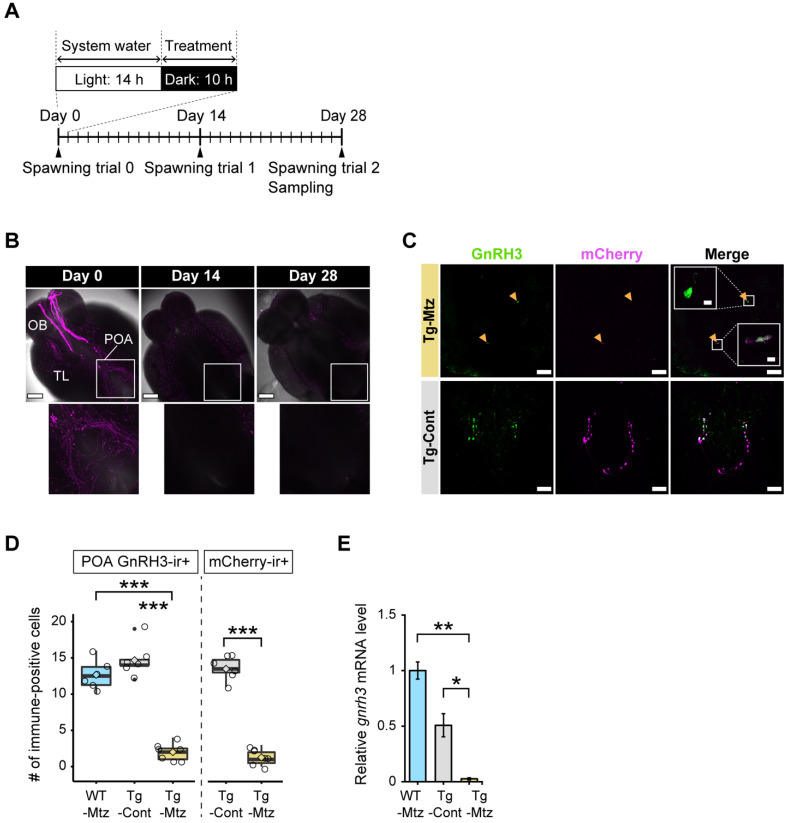
GnRH3 neuronal ablation is validated in the brain at 28 days of treatment. (**A**) Treatment and analysis schedule. Fish were exposed to 2 mM Mtz daily during the dark phase for 28 days, and were sampled at 14 and 28 days into the treatment. (**B**) Top panel: confocal images of GnRH3 neurons expressing NTR-mCherry fusion protein (magenta) in whole brains from the *Tg(gnrh3:Gal4ff; UAS:nfsb-mCherry)* fish before the treatment, and at 14 and 28 days of treatment. Bottom panel: POA-zoomed images showing the boxed area from the top images. Scale bars, 200 µm. (**C**) Confocal images of co-immunostaining of GnRH3-associated peptide (green) and mCherry (magenta) in the whole brain of Tg-Mtz (top panels) and Tg-Cont (bottom panels) at 28 days of treatment. GnRH3 neuron cell bodies co-localized with mCherry in Tg-Mtz are labeled by arrowheads. Scale bars, 100 µm. Insets, zoomed images of GnRH3-ir and mCherry-ir neuron cell bodies co-localization showing the boxed areas of the merged image in Tg-Mtz. Scale bars, 10 µm. (**D**) Numbers of GnRH3-positive cells (left) and mCherry-positive cells (right) in the POA (*n* = 6–7). (**E**) Relative *gnrh3* mRNA levels in the whole brain (*n* = 4 each). OB, olfactory bulb; TL, telencephalon; POA, preoptic area; ir, immunoreactive; WT-Mtz, WT siblings treated with 2 mM metronidazole (Mtz); Tg-Cont, *Tg(gnrh3:Gal4ff; UAS:nfsb-mCherry)* treated with vehicle; Tg-Mtz, *Tg(gnrh3:Gal4ff; UAS:nfsb-mCherry)* treated with 2 mM Mtz. *, *p* < 0.05; **, *p* < 0.01; ***, *p* < 0.001.

**Figure 2 ijms-23-05596-f002:**
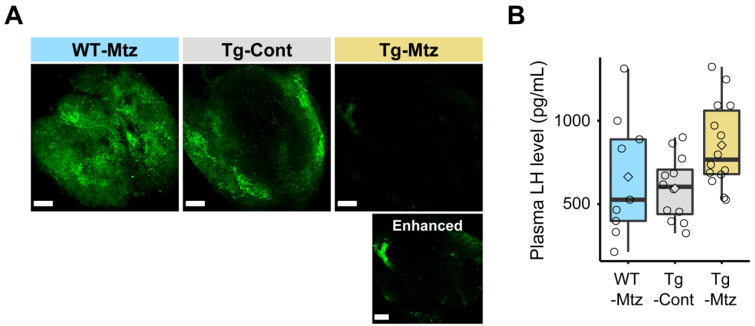
GnRH3 neuronal projections are reduced at 28 days of treatment in the pituitary of GnRH3 neuron-ablated females. (**A**) Confocal images of immunostaining with GnRH3-associated peptide antibody in the whole pituitary (posterior to top **left**), showing GnRH3 neuronal projections (green). Scale bars, 1 mm. The bottom panel of Tg-Mtz, an image with enhanced signals with levels and tone curve adjustments to demonstrate that GnRH3 is still detectable in the pituitary. (**B**) LH levels in plasma (*n* = 9–14). WT-Mtz, WT siblings treated with 2 mM metronidazole (Mtz); Tg-Cont, *Tg(gnrh3:Gal4ff; UAS:nfsb-mCherry)* treated with vehicle; Tg-Mtz, *Tg(gnrh3:Gal4ff; UAS:nfsb-mCherry)* treated with 2 mM Mtz. *p* > 0.05.

**Figure 3 ijms-23-05596-f003:**
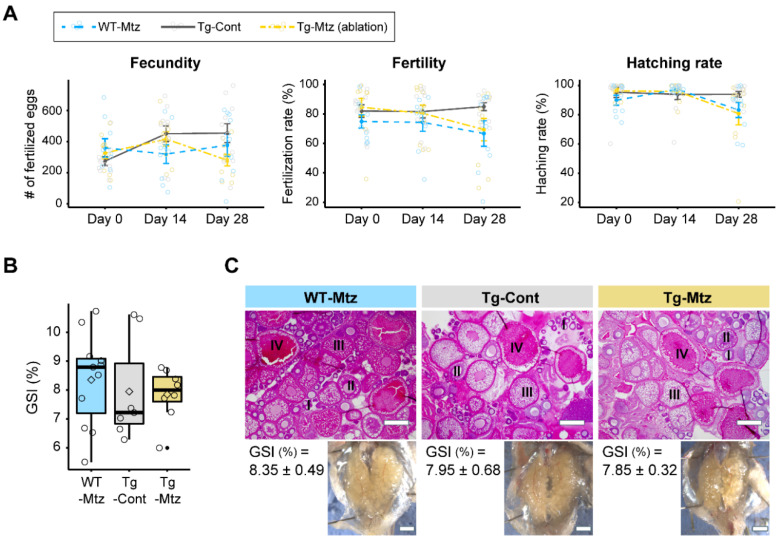
GnRH3 neuron-ablated females exhibit normal fertility. (**A**) Number of fertilized eggs per spawn (fecundity), percentage of fertilized eggs (fertility), and hatching rate from the experimental females paired with WT males before treatment, and at 28 days of treatment. (**B**) Gonadosomatic index (GSI) (*n* = 7–11). (**C**) Representative ovarian sections stained with hematoxylin and eosin. Scale bars, 500 µm. Bottom panels, images of the intact ovaries with the corresponding GSI (mean ± SEM) adjacent to each image. Scale bars, 1 mm. WT-Mtz, WT siblings treated with 2 mM metronidazole (Mtz); Tg-Cont, *Tg(gnrh3:Gal4ff; UAS:nfsb-mCherry)* treated with vehicle; Tg-Mtz, *Tg(gnrh3:Gal4ff; UAS:nfsb-mCherry)* treated with 2 mM Mtz. *p* > 0.05.

**Figure 4 ijms-23-05596-f004:**
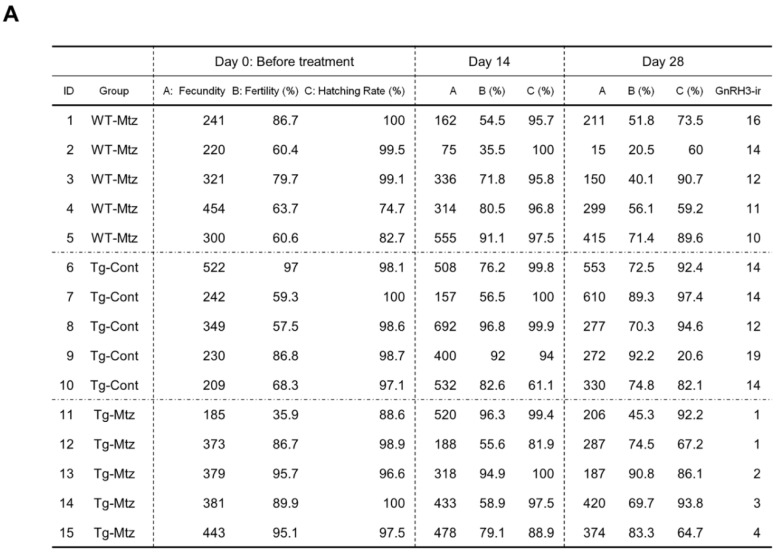
No correlation is observed between the number of detectable POA GnRH3 neurons and each reproductive parameter at 28 days of treatment. (**A**) Table depicting the number of remaining detectable GnRH3 immunoreactive (ir) neurons in the preoptic area, and the results of reproductive performance test for individual females from 0 days (before treatment) to 28 days of treatment. (**B**) Pearson’s correlation coefficient analysis between the number of POA GnRH3-ir cells and each reproductive parameter at 28 days of treatment. *R*, correlation coefficient. *P*, *p*-value. Data from A. WT-Mtz, WT siblings treated with 2 mM metronidazole (Mtz); Tg-Cont, *Tg(gnrh3:Gal4ff; UAS:nfsb-mCherry)* treated with vehicle; Tg-Mtz, *Tg(gnrh3:Gal4ff; UAS:nfsb-mCherry)* treated with 2 mM Mtz.

**Figure 5 ijms-23-05596-f005:**
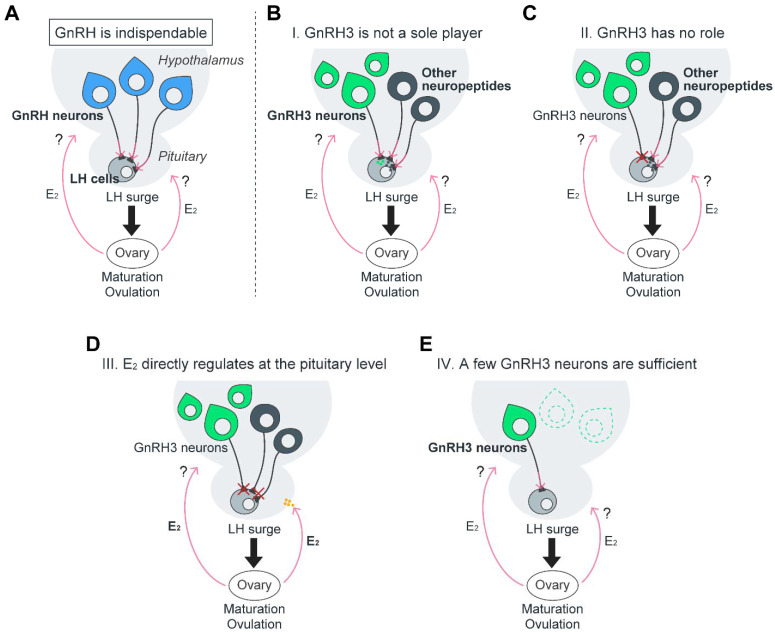
Schematic diagram of the different possible pathways in inducing the pre-ovulatory LH surge along the HPG axis in female zebrafish. In general, the accepted dogma is that the hypophysiotropic GnRH and its neurons are critical for LH surge, and ultimately oocyte maturation, ovulation, and spawning (**A**). Our current results suggest that GnRH3 neuron-ablated zebrafish females display normal fertility, and previous studies suggest four alternative pathways (**B**–**E**): I. Hypophysiotropic GnRH3 has no role, but other brain factors (e.g., vasoactive intestinal peptide, kisspeptin, neurokinin B, and secretoneurin) trigger LH surge (**B**); II. Functional redundancy—GnRH3 has a role, but is not the sole and only player, thus sharing this function with other factors (**C**); III. Ovarian estradiol-17*b* (E_2_) directly induces the LH surge and controls ovulation (**D**); IV. In GnRH3 neuron-ablated females, a few remaining hypophysiotropic GnRH3 neurons are possibly sufficient to elicit LH surge (**E**). The question marks indicate that these pathways remain to be established in zebrafish.

## Data Availability

The data presented in this study are available in this published article and in Appendix A, reposited in Figshare at https://doi.org/10.6084/m9.figshare.19662762 (accessed on 27 April 2022).

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
