# Peer review of "Chemogenetic Depletion of Hypophysiotropic GnRH Neurons Does Not Affect Fertility in Mature Female Zebrafish"

_ijms, 2022, doi:10.3390/ijms23105596_

Round 1

Reviewer 1 Report

The manuscript reports the effect of Chemogenetic Depletion of Hypophysiotropic GnRH3 Neurons in mature female zebrafish. There are studies showing that depletion of GnRH3 in zebrafish has no effect on fertility and ovulation. However, laser ablation of GnRH3 neurons at 4- and 6-days post-fertilization (dpf) resulted in arrested oocyte development and infertility in female zebrafish, implicating GnRH3 neurons, rather than GnRH3 peptide, in female reproduction. It is interesting to see which is the role of GnRH3 and to confirm the hypothesis that GnRH3 neurons are more involved than GnRH3 peptides in female reproduction in zebrafish. This article contributes to increase this knowledge by providing evidence that the hypophysiotropic GnRH and its neurons are likely dispensable for the regulation of maturation, ovulation, and spawning in female zebrafish. However many doubts remain about the mechanism that regulates the Lh surge and ovulation in female zebrafish.

The experimental methods were meticulous and sound. The presentation of the results in the manuscript is clear and concise.

Overall, the conclusion are well supported by the results.

Reviewer 2 Report

The manuscript entitiled: "Chemogenetic depletion of hypophysiotropic GnRH neurons does not affect fertility in mature female zebrafish" has been evaluated carefully.  In this very interesting article the authors provide new insights about GnRH and its neurons with regards to the LH surge in zebrafish, more precisely, about the relevance of hypophysiotropic GnRH1 and GnRH3 for the ovulation. The study is well designed and the artilce is written in a very clear way.

However, his reviewer has minor comments which need to be adressed.

1) The authors should eleaborate in the discussion a bit more on Kisspeptin, recent studies about the kisspept pathway and connections to GnRH, and relevance for onset of puberty and further fertility. 

2) The authors did not indicate the age of females which were used for the experiments. Please fix this issue. With regards to this, please briefly discuss the effects of aging/brain aging/ reproductive aging on the hypothalamus-pituitary-gonadal-axis.
